# Influence of Hydrated Lime on the Chloride-Induced Reinforcement Corrosion in Eco-Efficient Concretes Made with High-Volume Fly Ash

**DOI:** 10.3390/ma13225135

**Published:** 2020-11-14

**Authors:** Manuel Valcuende, Rafael Calabuig, Ana Martínez-Ibernón, Juan Soto

**Affiliations:** 1Department of Architectural Constructions, Universitat Politècnica de València, Camino de Vera, s/n., 46022 Valencia, Spain; rcalabui@csa.upv.es; 2Research Center for Molecular Recognition and Technological Development (IDM), Universitat Politècnica de València, Camino de Vera, s/n., 46022 Valencia, Spain; anmarib@arqt.upv.es (A.M.-I.); juansoto@qim.upv.es (J.S.)

**Keywords:** concrete, hydrated lime, fly ash, chloride, resistivity, corrosion rate

## Abstract

The main objective of this study was to analyze the influence that the addition of finely ground hydrated lime has on chloride-induced reinforcement corrosion in eco-efficient concrete made with 50% cement replacement by fly ash. Six tests were carried out: mercury intrusion porosimetry, chloride migration, accelerated chloride penetration, electrical resistivity, and corrosion rate. The results show that the addition of 10–20% of lime to fly ash concrete did not affect its resistance to chloride penetration. However, the cementitious matrix density is increased by the pozzolanic reaction between the fly ash and added lime. As a result, the porosity and the electrical resistivity improved (of the order of 10% and 40%, respectively), giving rise to a lower corrosion rate (i_CORR_) of the rebars and, therefore, an increase in durability. In fact, after subjecting specimens to wetting–drying cycles in a 0.5 M sodium chloride solution for 630 days, corrosion is considered negligible in fly ash concrete with 10% or 20% lime (i_CORR_ less than 0.2 µA/cm^2^), while in fly ash concrete without lime, corrosion was low (i_CORR_ of the order of 0.3 µA/cm^2^) and in the reference concrete made with Portland cement, only the corrosion was high (i_CORR_ between 2 and 3 µA/cm^2^).

## 1. Introduction

Due to its increasing concern regarding environmental issues, the concrete industry is trying to reduce the consumption of natural resources by reusing waste such as recycled construction aggregates or metallurgical residues [1,2], some of which have cementitious properties, for example fly ash [3]. This may be used as a partial cement replacement in concrete, making construction more sustainable by reducing CO_2_ emissions into the atmosphere (less cement is consumed) and waste management more efficient.

Fly ash is an artificial pozzolana that initially acts as an inert filler, filling the capillary pores. The pozzolanic reaction largely depends on temperature; according to Hanehara et al. [4], when concrete is cured at 20 °C, the fly ash reaction begins at the age of 28 days, while at 40 °C, the reaction has already started at 7 days. At early ages, this material also accelerates cement reactions due to the fineness of its particles, since it provides additional surfaces for nucleation (nucleation sites) of the hydration products [5]. Its hydration is much slower than Portland cement and the hydrates that form fill the existing pores, causing long-term refinement of the porous structure [6]. Moreover, according to Simičič et al. [7], fly ash particles modify the pore shape and give rise to lower effective porosity. From the point of view of durability, chloride binding capacity is another advantage that this material provides [8]. Chlorides may react with the alumina in the fly ash to form calcium chloro-aluminates, and therefore, the amount of free chlorides in the concrete decreases [8]. Furthermore, the fly ash pozzolanic reaction reduces the pore solution pH (less portlandite content) and modifies the chloride binding mechanisms: the lower OH^−^ concentration promotes Cl^−^ absorption on the surface of hydration products (chloride ions compete with the hydroxide ions for the adsorption sites) and raises the number of bound chlorides [9]. As a result, chloride transport slows down and the risk of corrosion decreases because only free chlorides can produce steel depassivation [10,11].

When fly ash is used as a partial cement replacement, not as an addition, the early strength of fly ash concrete is reduced due to the slow pozzolanic reaction of the ash [12,13]. Nevertheless, the replacement of cement with up to 30–35% fly ash improves certain properties of concrete in both fresh and hardened states (higher workability, lower hydration heat, and higher strength and durability in the long-term) [3,14,15,16,17,18], so this percentage is generally considered optimal and is often used. In fact, cement containing up to 35% fly ash (CEM II/B-V) is made in Europe and some European standards, such as Spanish standard EHE-08, allow the addition of this percentage of fly ash to reinforced concrete when cement without additions is used. Resistance to chloride penetration and oxygen permeability also improve [7,12], making such concretes appropriate for use in marine environments [19], even though early porosity is high. However, at higher cement replacement percentages (over 40–45%), the mechanical properties and elastic modulus tend to be reduced [14,20,21,22,23,24], with a further reduction with increased fly ash content. For example, Huang et al. [22] reported a 48% and 35% compressive strength drop in cement pastes with 50% fly ash replacing cement, at 7 and 28 days, respectively. Anjos et al. [23] recorded slightly higher strength decreases in concrete. Likewise, according to Lorca et al. [25], concrete with 50% replacement of cement by fly ash showed a 50%, 46%, and 27% compressive strength reduction at 7, 28, and 365 days compared to reference concrete, while with a 75% replacement, the reduction was 70%, 60%, and 45%. Alaka and Oyedele [21] also pointed out decreases in flexural and splitting tensile strength in concretes with 60% and 65% cement replaced by fly ash compared to concrete with 50% replacement. Durability may be also compromised because the porosity and water absorption increase [22,26], the carbonation rate is higher [24,26,27,28,29], and the Ca(OH)_2_ required for the pozzolanic reaction of fly ash may be insufficient and may reduce the pH [4,23,25,30,31]. Concrete with least 50% of the cement replaced by fly ash is known as high volume fly ash (HVFA) concrete.

Lime increases the availability of portlandite in concrete [25,32,33], thus favoring and accelerating fly ash hydration [34]. However, lime contribution depends largely on the amount of fly ash and cement, being of little importance in concrete mixes with a low reduction of Portland cement (15–25%) and higher with replacements of 50% [25]. Therefore, the addition of small amounts of hydrated lime in HVFA concretes tends to compensate the compressive strength loss that occurs in these concretes due to the high substitution of cement by fly ash. For example, Lorca et al. [25] made concretes with 50% cement replacement by fly ash plus 20% lime and obtained equivalent strength to concrete made with 100% Portland cement. However, for higher cement replacements equal to or higher than 60%, the compressive strength loss is very important, and the addition of lime cannot compensate this strength loss [23,25,35]. That is why, in this research, the durability study has been carried out on concretes with 50% cement replacement, since to obtain sustainable concrete, it is important to reduce the consumption of Portland cement but maintain its mechanical and durability properties.

To date, studies of HVFA concrete with added lime have been scarce and have focused on analyzing their mechanical properties and microstructure [23,25,32,36]. In order to increase the knowledge and study the durability of this concrete in aggressive environments, such as the marine environment, the aim of this research was to analyze the influence of hydrated lime addition on the chloride-induced reinforcement corrosion in concrete made with 50% cement replacement by fly ash. The corrosion rate was determined by the polarization resistance method. Since the kinetics of corrosion depend on several factors, other tests such as mercury intrusion porosimetry, chloride penetration, and electrical resistance were also carried out in order to explain the behavior observed. The influence of lime addition on the compressive strength was also studied, and the results were compared to those from a reference concrete made with 100% Portland cement.

## 2. Experimental

### 2.1. Concrete Mixtures and Materials

Four different concrete mixes were made with w/b (water/binder) ratio of 0.5: a reference concrete made with Portland cement only (C-0) and three mixes with 50% of the cement replaced by fly ash. Varying amounts of hydrated lime were also added to the three mixes: the first with 0% by weight of fly ash (CFA-0L), the second with 10% (CFA-10L), and the third with 20% (CFA-20L).The lime content was limited to a maximum of 20%, because with this content, Lorca et al. [25] obtained the best results. CEM I 52.5 R cement and two types of limestone aggregates, gravel 4/8 and sand 0/4, were used. The fly ash corresponds to a V type fly ash according to standard EN 197-1:2011. The superplasticizer content, which was a polycarboxylate-based admixture (Sika Viscocrete 3425), was adjusted to achieve a slump of 150 mm ± 10 mm in the Abrams cone for the different mixtures. Mixtures made with cement and fly ash required less superplasticizer than mixtures made with cement only, and mixtures made with added lime required more superplasticizer. The characteristics of each mix are shown in Table 1, and the chemical composition of cement and fly ash are shown in Table 2.

The hydrated lime used was CL 90 S to standard EN 459-1:2011, with a purity of 92% in Ca(OH)_2_, a density of 2.3 g/cm^3^, a particle size distribution similar to the fly ash, and with an average particle size of 22.8 μm. The grading curves of the fly ash and lime were determined by laser diffraction (Figure 1). The chemical composition of lime is shown in Table 3.

Setting time was also measured in concrete mixes (ASTM C 403-08), to obtain further information on the influence of replacing cement by fly ash and adding lime (Figure 2). The 50% cement replacement by fly ash quite delays the setting time, because of the decrease in Portland cement content of the binder system [26,37]. However, the addition of lime to fly ash concrete accelerates the setting time by more with 20% than 10% lime. The initial and final setting is practically the same in reference concrete (C-0) and in fly ash concrete with 10% lime (CFA-10L) but lower with 20% lime (CFA-20L). These results agree with those obtained by Bentz [37], who points out the ability of lime to accelerate and amplify the cement hydration reactions.

### 2.2. Test Program and Methodology

Compressive strength was tested according to European standard EN 12390-3. Tests were made at five different ages: 7, 28, 90, 180 and 365 days. Three batches were made from each mix, and two samples from each batch were tested at each age.

To analyze the extent of chloride corrosion in the different concretes, six types of measurements were performed: mercury intrusion porosimetry, chloride migration coefficient, accelerated chloride penetration test, electrical resistivity, and corrosion rate. Electrical conductance tests were also carried out on different solutions of lime and fly ash in order to analyze the influence of adding lime on pore water conductivity.

Different types of specimens were used and were removed from their molds 24 h after casting. Then, they were placed in the curing chamber at 20 °C and above 95% RH until the age of 28 days. As the pozzolanic reaction of fly ash is quite slow and only occurs in the presence of water, the specimens were subsequently immersed in water at 20 °C until the tests started.

#### 2.2.1. Mercury Intrusion Porosimetry (MIP) Test

At 28 and 180 days, pore size distribution was determined by a Micromeritics AutoPore IV-9500 mercury porosimeter (Micromeritics GmbH, Mönchengladbach, Germany). The test was carried out on small drilled cores (12 mm diameter × 23 mm high) weighing approximately 6 g. The cored samples were obtained from the center of 100 × 100 × 100 mm cubic specimens. They were first dried in an oven (Memmert GmbH+Co.Kg, Schwabach, Germany) at 105 °C and then immersed in mercury under gradually increasing pressure. This technique can also measure total porosity of the sample.

#### 2.2.2. Chloride Migration Coefficient Test

This test, performed to the NT BUILD 492 standard, provides a measure of the resistance of the concrete to chloride penetration. Cylindrical specimens with a diameter of 100 mm and a thickness of 50 mm were used. An external electrical potential was applied axially across the specimen to force the chloride ions outside to migrate into the specimen. At the time of the test, the specimen was axially split, and a silver nitrate solution was sprayed on the freshly split sections. Then, the chloride penetration depth could be measured from the visible white silver chloride precipitation, subsequently allowing the chloride migration coefficient to be calculated. The test was carried out at the age of 90 days. Three batches were made from each mix, and two samples from each batch were tested. The result was the arithmetic mean of all six values.

#### 2.2.3. Accelerated Chloride Penetration Test

To perform this test, 40 × 40 × 160 mm prismatic specimens were submerged in water until the age of 100 days and then exposed to successive wetting–drying cycles in a 0.5 M sodium chloride solution. The chloride concentration used is similar to that found in seawater. The time for each cycle was 7 days; the specimens were immersed in the solution for 4 days, then dried in an oven at 50 °C for 2 days, and finally cooled at room temperature for 1 day. The chloride penetration depth was measured at three different ages: after 3, 9, and 14 cycles (21, 63, and 98 days, respectively). To perform the measurements, the specimens were axially split, and a silver nitrate solution was sprayed on the freshly split sections. Three batches were made from each mix, and two samples from each batch were tested. The result was the arithmetic mean of all six values.

#### 2.2.4. Electrical Resistance Test

Electrical resistance was determined by the reference method (direct method) by applying a uniform electric field using two electrodes in contact with the specimen bases [38]. Prismatic specimens with a squared section 160 mm long and 40 mm wide were made and stored underwater at 20 °C during the test period. The test was carried out as described in Gandía-Romero et al. [39]. Three batches were made from each mix, and two samples from each batch were tested. The result was the arithmetic mean of all six values.

#### 2.2.5. Corrosion Rate Test

Cylindrical specimens 50 mm in diameter and 100 mm long with a 12-mm diameter bar (steel B 500 SD) in the center were made. The bar protruded through one side of the specimen (Figure 3). The concrete cover of the bar was 18 mm. Following the UNE 112072:2011 standard guidelines [40], the bar was covered with insulating tape, the exposed steel area being 2262 mm^2^. The electrical contact for measurements was at the end of the bar, which was protected with Vaseline in order to prevent corrosion. After casting, the specimens were cured for 28 days and remained submerged in water until the age of 100 days. Then, the specimens were subjected to 7-day wetting–drying cycles in a 0.5 M sodium chloride solution (as described in Section 2.2.3) for 630 days (until the age of 730 days). Later, the specimens were axially split for a visual estimation of rebar corrosion and then cleaned firstly by brush and then with phosphoric acid in three 10-min cycles.

Corrosion current density (i_CORR_) was measured by the linear polarization resistance method [40] on an Autolab/PGSTAT128N potentiostat (Metrohm, Utrecht, The Netherlands). Once the polarization resistance (R_P_) was obtained, and considering the ohmic drop between the bar and the reference electrode, i_CORR_ was determined, i_CORR_ = B/(A·R_p_), where B is the Stern–Geary constant, which is assumed to be 26 mV [40,41], and A is the exposed steel area. Three batches were made from each mix, and two samples from each batch were tested. The result was the arithmetic mean of all six values.

#### 2.2.6. Electrical Conductance Test in Lime and Fly Ash Solutions

To apply this method, different mass percentages of lime/fly ash were used (0.0, 0.025, 0.05, 0.1, 0.2, and 0.6). The total solid mass prepared for each mix was 2 g. First, a beaker was filled with 100 mL of distilled water, hermetically closed to prevent water loss by evaporation, and placed in a thermostated cell until the test temperature of 60 °C was reached. This temperature was chosen to speed up the reactions. Subsequently, the lime was added and conductance measured; then, fly ash was put inside, and conductance readings were obtained at different ages. The beakers were kept at a constant temperature of 60 °C ± 0.1 °C and under argon to avoid carbonation for the whole period. Likewise, the solutions were stirred continuously to facilitate the pozzolanic reaction (Figure 4). The conductance measurements were performed with a commercial conductimeter (Crison GLP32, Crison instruments, Barcelona, Spain).

## 3. Results and Discussion

### 3.1. Effect of the Lime on Compressive Strength

Figure 5 shows the evolution of compressive strength up to 360 days of concrete. The difference between the reference concrete (C-0) and the one with a 50% replacement of cement by fly ash (CFA-0L) is considerable at 7 days (around 45% less strength) and becomes significantly smaller after 3 months. This can be considered normal in this type of binder, since the benefits of the pozzolanic reaction occur in the mid and long term. In fact, CFA-0L has, on average, a compressive strength 21% and 15% lower than the reference concrete at 180 and 360 days, respectively.

In concrete mixes with fly ash (CFA-0L, CFA-10L, CFA-20L), adding lime to the mixture improves the mechanical strength with the best performance at 20% lime. The increased strength is not significant at 7 days. At older ages, the added lime participates in the fly ash pozzolanic reaction, giving rise to new calcium silicate hydrates and therefore obtaining higher compressive strengths than in the concrete without lime. For example, at 1 year, the strength of CFA-10L and CFA-20L is respectively 11% and 18% higher than CFA-0L. At the same age, this increase in strength also enabled concrete mixes with 50% less cement and 20% added lime to be similar in strength to the reference concrete (C-0). These results agree with those obtained by Lorca et al. [25] and show that the fly ash has reacted in a significant way with the alkaline addition.

### 3.2. Mercury Intrusion Porosimetry (MIP)

Figure 6 and Figure 7 show the mercury intrusion volume according to the equivalent pore diameter. As expected, porosity decreases with age and the porous structure becomes finer due to the progressive formation of C-S-H.

At the two ages studied, concretes with a 50% cement replacement by fly ash (CFA-0L, CFA-10L, CFA-20L) have higher total porosity than the concrete made with Portland cement only (C-0) (Table 4). These results agree with those obtained from the compression tests, in which, for example, at 180 days compressive strength in C-0 is 26%, 14%, and 5% higher than CF-0L, CF-10L, and CF-20L, respectively, and the total porosity is 22%, 13%, and 10% lower, respectively.

As it can be seen in the evolution of compressive strength, in C-0, the decrease in porosity from 28 to 180 days is very little (from 9.4% to 9.0%), which is probably due to the use of a high early strength cement and the absence of pozzolanic reaction. On the contrary, the porosity of fly ash concretes is reduced even more in the same period, matching the findings of other research groups [5,42], and it is due to the denser cementitious matrix produced by the fly ash pozzolanic reaction, which reduced the capillary pore volume and interconnectivity between pores. As a result, at 180 days, the pore distribution is similar in all four concrete mixes and the fineness of the porous structure tends to be the same. In fact, at this age, the maximum pore concentration is found in all concretes for pore sizes of 0.02–0.03 μm. Moreover, the pore size for rapid mercury intrusion is practically the same in all four concretes, around 0.06–0.08 μm. This diameter, known as the threshold diameter, signals the limit from which the highest number of pores is concentrated and so is a good indicator of pore structure fineness.

Regarding only the influence of hydrated lime on fly ash concrete mixes (CFA-0L, CFA-10L, and CFA-20L), the total pore volume is lower in concretes with added lime and is lowest with an addition of 20% (Table 4). For example, at 180 days, porosity in CFA-0L, CFA-10L, and CFA-20L is 11.5%, 10.3%, and 10.0%, respectively. Furthermore, the reduction of porosity occurred between 28 and 180 days is greater in CFA-10L and CFA-20L concretes, indicating higher pozzolanic activity (generating more C-S-H) and, therefore, an increase in durability. These results agree again with those obtained from the compression tests and explain the better compressive strength of CFA-20L compared with CFA-10L and CFA-0.

### 3.3. Resistance to Chloride Penetration

Figure 8 shows the chloride migration coefficient (D_nssm_) obtained at the age of 90 days. The chloride penetration for concretes made with 50% cement replaced by fly ash (CFA-0L, CFA-10L, and CFA-20L) is much lower than concrete without fly ash (mix C-0), giving D_nssm_ values of the order of 10 times lower. Differences in chloride permeability were also observed in the accelerated chloride penetration test (Figure 9); for example, after exposing the specimens to cycles of wetting–drying for three weeks in a 0.5 M sodium chloride solution, the chloride penetration depth for fly ash concretes was between 1 and 2 mm, and in concrete without fly ash, the depth was 10 mm on average. These results are consistent with those obtained in other research works [7], in which concretes with 47% higher total porosity nevertheless have higher resistance to chloride penetration. This is due to the high alumina content of fly ash (Table 2), since chlorides react chemically with tricalcium aluminate or its hydrates to form calcium chloro-aluminate (Friedel’s salt) [43]. According to Thomas et al. [8], the quantity of binding generally increases with the amount of alumina, and even if there is a decrease in the free chloride content, due for example to leaching, part of the chlorides remains bound.

Regarding only the influence of lime on the behavior of concretes made with fly ash (CFA-0L, CFA-10L, and CFA-20L), Figure 8 shows that the D_nssm_ coefficient of concretes with lime (CFA-10L and CFA-20L) are similar to those for concrete without lime (CFA-0L). The chloride penetration depth in the specimens subjected to wetting–drying cycles is also similar, although it tends to be slightly higher in concretes with 10% and 20% of lime. As seen in Figure 9, the differences between concrete CFA-0L and concretes CFA-10L and CFA-20L are on average only 1 mm after three months. In any case, adding small amounts of finely ground hydrated lime to concretes with fly ash has a minimal effect on its permeability to chlorides.

### 3.4. Electrical Resistivity

Electrical resistivity is a property that reflects the ability of a material to transport electric charge. In concrete, charge is transported through the ions dissolved in the pore solution, and therefore, the resistance provides an indication of pore connectivity. This parameter is also related to corrosion [44], since a potential difference is established between anodic and cathodic areas of the reinforcing steel, so the corrosion current depends on the existing resistance between them. In concrete with high electrical resistivity, corrosion is slow, because the current cannot easily pass between anodic and cathodic areas.

Figure 10 shows electrical resistivity evolution over time. As expected, the concrete resistivity increases with age due to the progressive formation of C-S-H, which densifies the microstructure and reduces the pore connectivity.

Concretes made with a 50% cement replacement by fly ash (CFA-0L, CFA-10L, and CFA-20L) have much higher resistivity than the reference concrete (C-0), even though the latter’s total porosity is lower (Table 4). For example, at 180 days, CFA-0L resistivity is around ten times higher than C-0. This is because resistivity depends on both pore structure (which reduces the mobility of ions inside the concrete) and the conductivity of the pore solution [10,45]. In fact, Zhang and Gjorv [46] found no direct relationship between water permeability and electrical conductivity.

Some additions are known to modify ion concentration in the pore solution [47,48,49,50]. According to Baroghel-Bouny et al. [12] or Shi [45], the conductivity of the pore solution mainly depends on the concentration of Na^+^, K^+^, and OH^−^; therefore, when this is reduced, conductivity also falls. Hussain and Rasheeduzzafar [51] point out that when the cement is substituted by fly ash, the OH^−^ concentration decreases. In fact, when replacing part of the cement with fly ash, there is a decrease in the calcium hydroxide content (promoted simultaneously by the cement reduction and pozzolanic reaction), which consequently decreases the pH of the pore solution and therefore the OH^−^ concentration, thus increasing the resistivity of concrete. Shi [45] or Yishun et al. [52] also found that the concentration of the alkali ions (Na^+^ and K^+^) generally decreases, leading to higher resistivity.

In certain cases, replacing cement by fly ash may increase conductivity, but according to Shehata et al. [53], this only happens with high-alkali fly ash (>5% Na_2_O_e_). This is not the case of the fly ash used in the present study, in which the total equivalent sodium oxide content (Na_2_O_e_) is less than 2.63% and with low lime and high silica content (Table 2). In short, replacing cement by fly ash considerably increases resistivity due to the lower conductivity of the pore solution (C-0 vs. CFA-0L).

Regarding the influence of hydrated lime on fly ash concretes (CFA-0L, CFA-10L, and CFA-20L), Figure 10 shows that resistivity rises with added lime; at 180 days, the rise is 46% and 31% in CFA-10L and CFA-20L, respectively. Contrary to this fact, the results obtained in the electrical conductance tests carried out on lime and fly ash solutions (Figure 11) show that lime initially tends to raise solution conductivity, that is, to decrease its resistivity. However, as may also be seen in Figure 11, when the added lime/fly ash ratio is small (e.g., 0.1, as in CFA-10L) the higher solution conductivity tends to disappear after a few weeks. Thus, adding small quantities of lime do not affect the pore water conductivity and, therefore, the lower conductivity (higher resistivity) of CFA-10L and CFA-20L with respect to CFA-0L is due to the cementitious matrix density being increased by the pozzolanic reaction between fly ash and added lime (discussed with MIP results).

Figure 11 also shows that when the added lime/fly ash ratio is 0.2 or higher (as in CFA-20L), the increase in the solution conductivity begins to be significant in the long term, which explains the lower resistivity found in CFA-20L than in CFA-10L.

To sum up, cement replacement by fly ash decreases the conductivity of the pore water, thus increasing the concrete resistivity, even though the total porosity may be higher. Furthermore, the addition of small quantities of hydrated lime does not significantly modify the pore water conductivity, but it does modify the concrete porosity, reducing it. In this case, it is the porosity that controls the mixture resistivity. As a consequence of all this, the resistivity of fly ash concretes with 10%–20% lime is higher than that of fly ash concrete without lime, and it is much higher than concrete made with Portland cement only.

### 3.5. Corrosion Rate

According to Spanish standard UNE 112072 [40], the reinforcement is considered to be in a passive state when i_CORR_ is lower than 0.1 µA/cm^2^ (negligible corrosion). From a practical point of view, some researchers establish the onset of corrosion when an average sustained corrosion rate higher than 0.2 µA/cm^2^ is reached [54,55].

Figure 12 shows the i_CORR_ evolution over time of rebars embedded in concrete specimens subjected to wetting–drying cycles in a 0.5 M sodium chloride solution. For all specimens, the i_CORR_ value was not negligible during the first days, and afterwards, it quickly dropped below 0.1 µA/cm^2^. Thus, a little corrosion occurs initially, but later, a passivating layer is created around the rebar, stabilizing the i_CORR_.

As shown in Figure 12, the corrosion rate in concretes made with a 50% cement replacement by fly ash (CA-0L, CA-10L, CA-20L) is much lower than those obtained in the reference concrete (C-0), even though the latter’s porosity is lower. For fly ash concretes, the i_CORR_ value was always lower than 0.35 µA/cm^2^, while for C-0 concrete, a significant increase in the corrosion rate occurred 100 days after starting the wetting–drying cycles, reaching moderate values of i_CORR_ (>0.5 µA/cm^2^) after 155 days and high values (>1.0 µA/cm^2^) after 210 days. Over 550 days, the corrosion rate tends to stabilize around values between 2 and 3 µA/cm^2^. These results are consistent with those reported in the accelerated chloride penetration test. As described in Section 3.3, after wetting–drying cycles for 63 days (Figure 9), the chloride penetration depth was greater than 20 mm and therefore higher than the concrete cover of the rebar, which was 18 mm. This means that the chlorides had reached the rebar, destroyed the passivating oxide layer and that corrosion had started. In fact, after splitting the specimens down the middle, there are clear signs of pitting corrosion (Figure 13).

This better behavior against the chloride corrosion of fly ash concretes is due to two reasons: (a) the resistance to chloride penetration is higher than C-0 because the fly ash combines with some of the free chlorides, reducing its diffusion (Section 3.3), and (b) the electrical resistivity of fly ash concretes is higher (Section 3.4), exerting an ohmic control on corrosion because corrosion kinetics depends, among other factors, on concrete resistivity [44,55].

Regarding only the influence of lime on fly ash concretes (CA-0L, CA-10L, and CA-20L), the addition of lime tends to slightly improve corrosion behavior. The corrosion rate was somewhat lower in CA-10L and CA-20L than CA-0L. For example, for concrete mixes with 10% and 20% added lime, the i_CORR_ values are very similar; they always stayed below 0.15 µA/cm^2^ and 0.17 µA/cm^2^, respectively, and there are no visible signs of rebar corrosion (Figure 14b). Conversely, in CA-0L after 450 days (350 days after starting the wetting–drying cycles), i_CORR_ is higher than 0.2 µA/cm^2^, which means that corrosion has started. In fact, the opened specimens have rust stains in the zone of the rebar ribs (Figure 14a). In this concrete, the total porosity (MIP test) is higher, and the electrical resistivity lower than CA-10L and CA-20L; thus, the current that circulates between cathode and anode inside the concrete is higher, increasing the corrosion rate.

To sum up, 50% cement replacement by fly ash increases the electrical resistivity of concrete and its resistance to chloride diffusion, reducing the probability of rebar corrosion. The addition of hydrated lime to these concretes also favors fly ash hydration and allows making up for, in the long term, the loss of compressive strength and the increase in porosity that occurs as a result of the high cement substitution by fly ash. This improvement is greater with 20% than 10% lime. In this way, the rebar protection against corrosion increases and hence the concrete durability. Therefore, the addition of 20% lime to fly ash concretes improves their mechanical properties and durability, obtaining more sustainable concrete.

## 4. Conclusions

The following conclusions can be drawn from the tests carried out on concretes with 50% cement replaced by fly ash:The addition of finely ground hydrated lime to concretes with 50% cement replacement by fly ash improved compressive strength, this increase being greater with 20% than 10% lime. The addition of 20% allowed a compressive strength similar to that of concrete made entirely with Portland cement to be obtained after one year, which means that the fly ash has reacted in a significant way with the alkaline addition.Adding lime to fly ash concrete reduced porosity, and to a larger extent with 20% rather than 10% added lime. At 180 days, this reduction was on average 10% and 13% in CFA-10L and CFA-20L concrete mixes, respectively, compared to CFA-0L. Still, all fly ash concretes were more porous than concrete with Portland cement only.In fly ash concretes, the chloride penetration depth was around 10 times lower than in concrete made with Portland cement, despite having higher total porosity. This is due to the high alumina content of fly ash. The addition of 10–20% of lime to fly ash concrete did not affect its permeability to chlorides.Fly ash concretes had around 10 times higher resistivity than the reference concrete at the age of 180 days. The addition of 10–20% lime to fly ash concrete does not significantly modify the pore water conductivity but increases the concrete resistivity due to a decrease in porosity.The corrosion rate in concrete with 50% cement replaced by fly ash was 85% lower than in the reference concrete due to higher resistance to chloride penetration and higher resistivity. Adding 10–20% lime to fly ash concrete slowed down the corrosion rate even more as a result of greater concrete resistivity. At 730 days, the i_CORR_ was of the order of 0.3 µA/cm^2^ in fly ash concrete without lime and less than 0.2 µA/cm^2^ in that with 10% or 20% lime.As a general conclusion, the addition of 20% lime to high-volume fly ash concrete increases its mechanical properties and the reinforcement protection against chloride-induced corrosion, obtaining more sustainable concrete.The results obtained make it interesting to continue studying the durability of these concretes, specially carbonation-induced reinforcement corrosion in HVFA concrete, since fly ash concrete has a lower carbonation resistance. This loss of resistance is also higher with increasing fly ash content.

## Figures and Tables

**Figure 1 materials-13-05135-f001:**
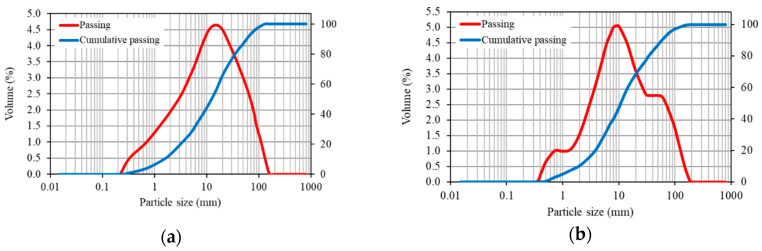
Grading curve: (**a**) fly ash; (**b**) lime.

**Figure 2 materials-13-05135-f002:**
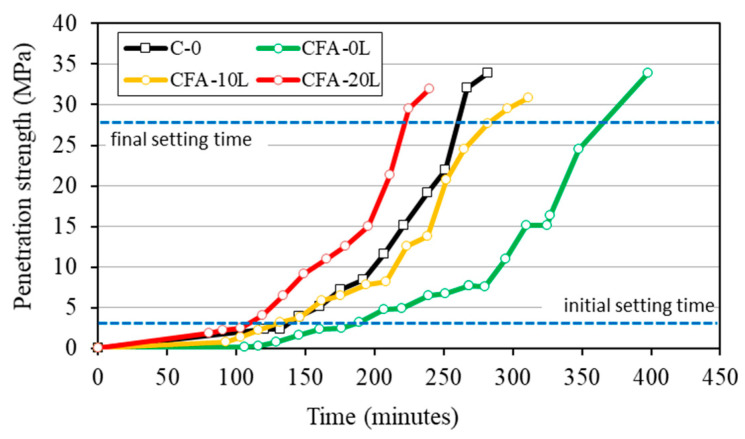
Setting time.

**Figure 3 materials-13-05135-f003:**
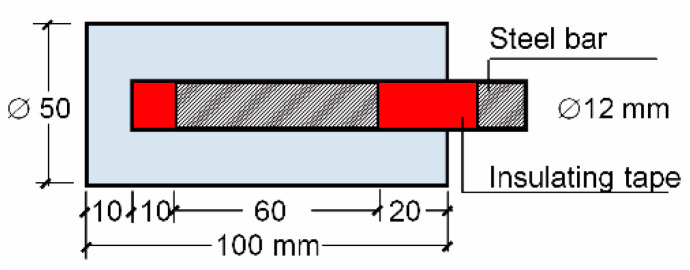
Longitudinal cross-section of the specimen.

**Figure 4 materials-13-05135-f004:**
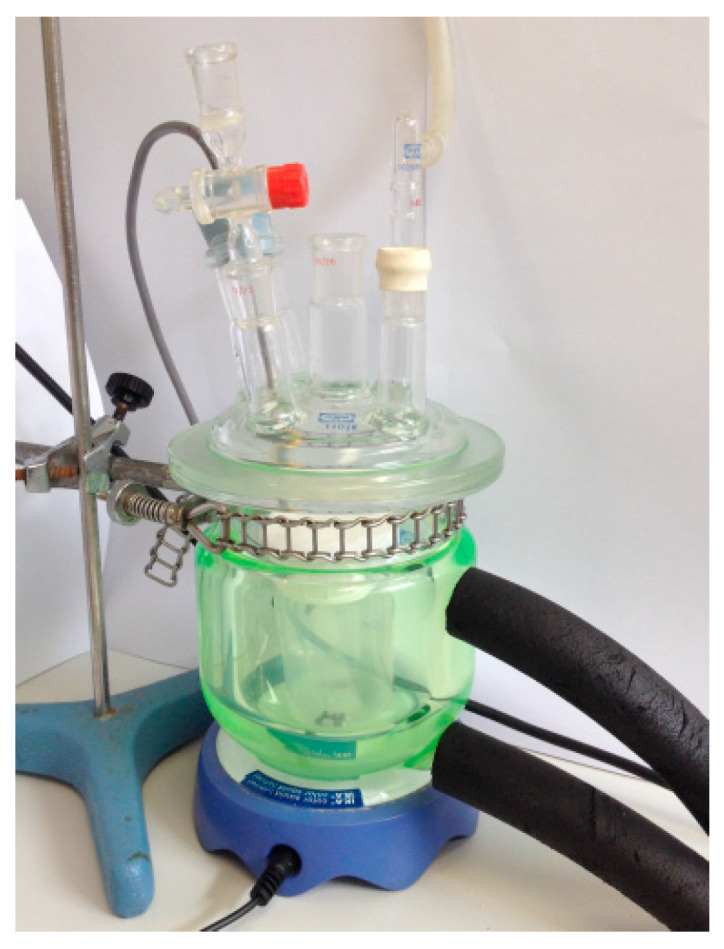
Thermostated cell for conductivity tests in solutions.

**Figure 5 materials-13-05135-f005:**
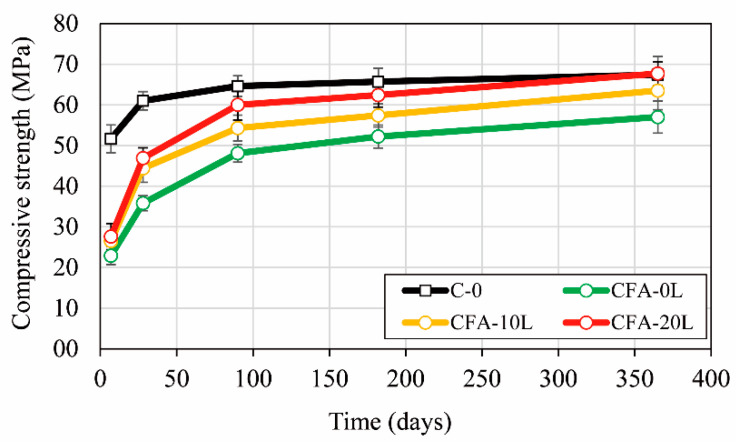
Evolution of concrete compressive strength.

**Figure 6 materials-13-05135-f006:**
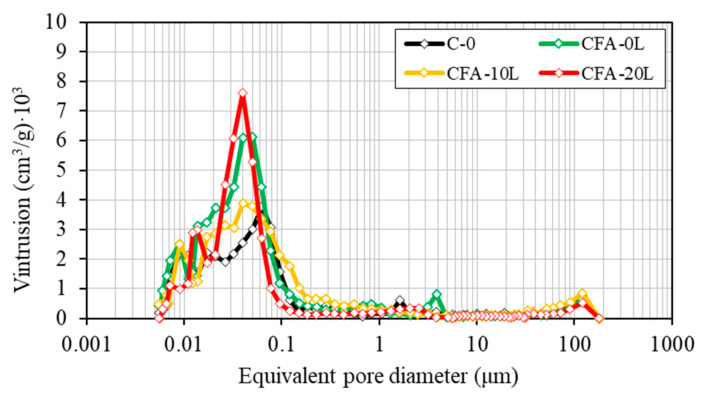
Pore size distribution at the age of 28 days.

**Figure 7 materials-13-05135-f007:**
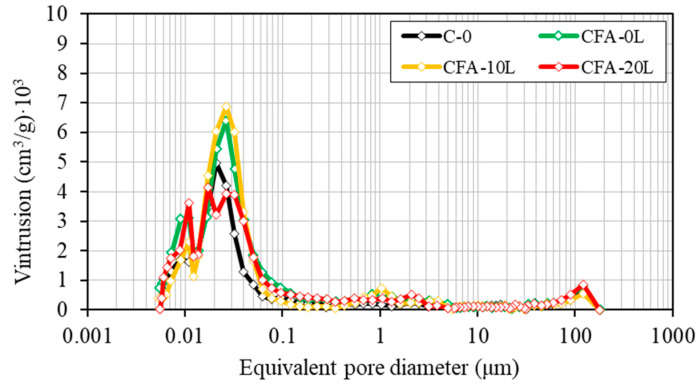
Pore size distribution at the age of 180 days.

**Figure 8 materials-13-05135-f008:**
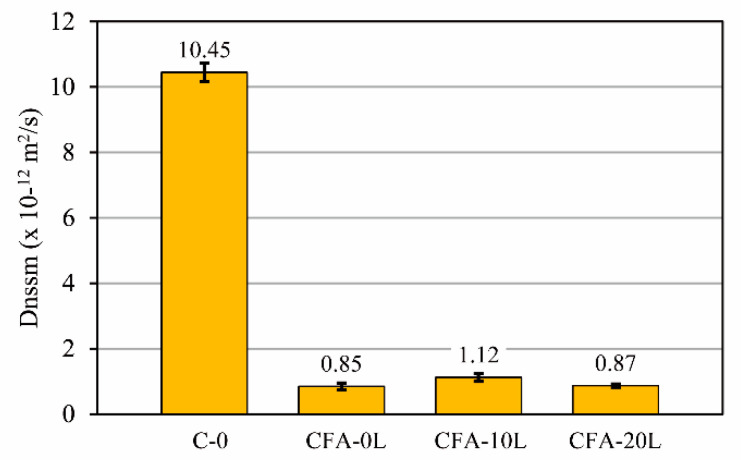
Chloride migration coefficient at the age of three months.

**Figure 9 materials-13-05135-f009:**
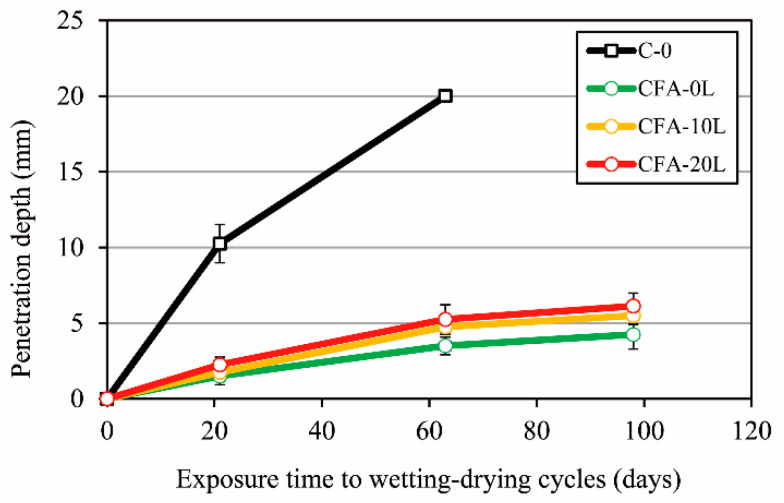
Chloride penetration depth in specimens subjected to wetting–drying cycles in a 0.5 M sodium chloride solution.

**Figure 10 materials-13-05135-f010:**
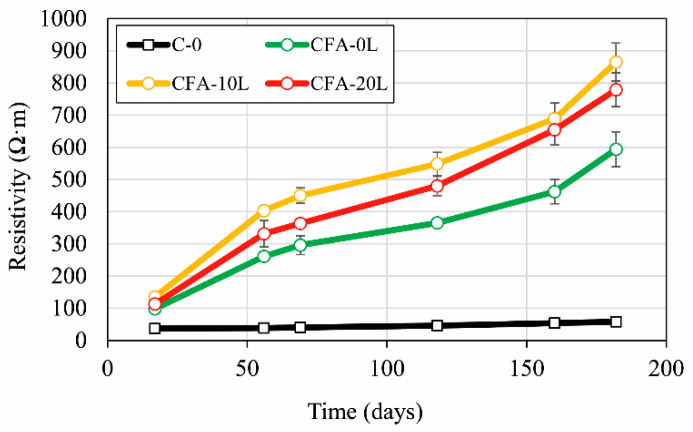
Evolution of concrete electrical resistivity.

**Figure 11 materials-13-05135-f011:**
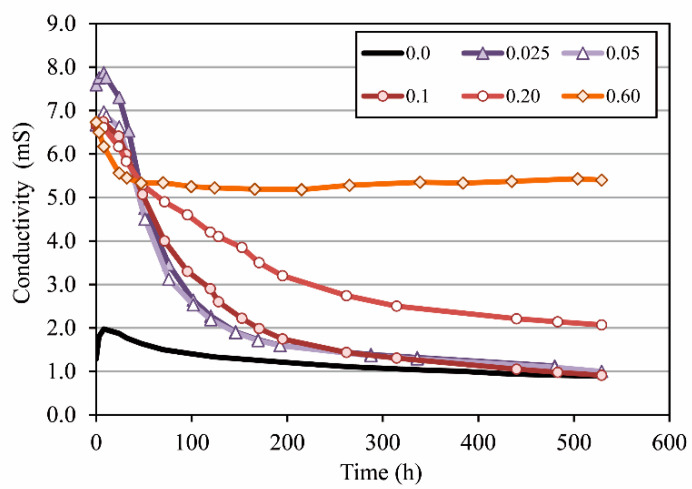
Electrical conductance test in solutions with different lime/fly ash ratios.

**Figure 12 materials-13-05135-f012:**
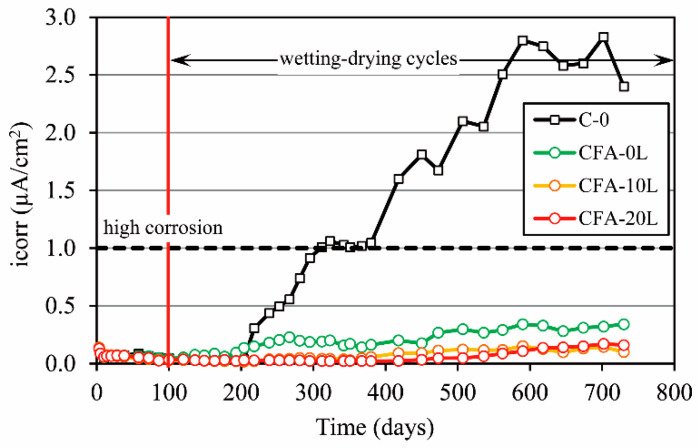
Evolution of the corrosion rate (i_corr_).

**Figure 13 materials-13-05135-f013:**
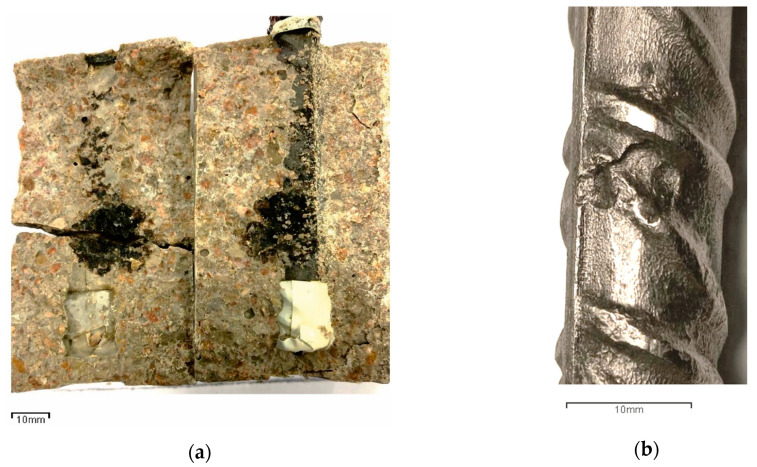
Rebar corrosion in a specimen of concrete C-0. (**a**) Specimen axially split; (**b**) pitting corrosion on the clean rust rebar.

**Figure 14 materials-13-05135-f014:**
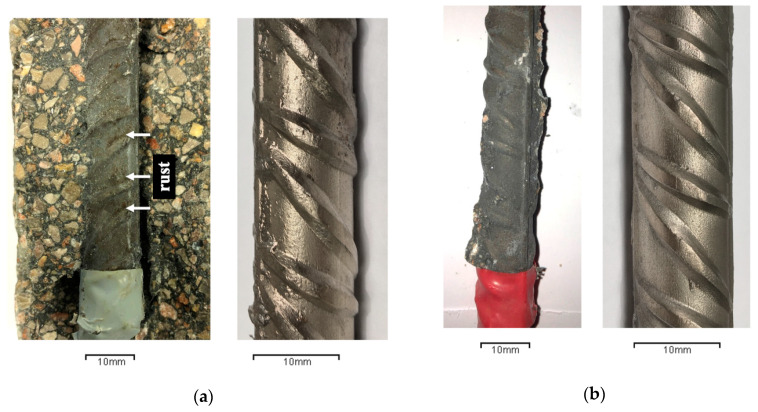
(**a**) Rebar in CFA-0L concrete; (**b**) rebar in CFA-20L concrete.

**Table 1 materials-13-05135-t001:** Mixture proportions of concretes and air content of fresh concrete.

Mix	Component							Air Content (%)
	Cement (kg/m^3^)	Water (L/m^3^)	w/b ^(^*^)^	Fly Ash (kg/m^3^)	Lime (kg/m^3^)	Gravel (kg/m^3^)	Sand (kg/m^3^)	
C-0	360	180	0.50	0	0	375	1500	4.2
CFA-0L	180	180	0.50	180	0	367	1467	3.3
CFA-10L	180	180	0.50	180	18	362	1450	3.7
CFA-20L	180	180	0.50	180	36	358	1433	4.4

^(^*^)^ binder: cement + fly ash.

**Table 2 materials-13-05135-t002:** Chemical composition of cement and fly ash, wt % (by X-ray fluorescence).

	SiO_2_	Al_2_O_3_	Fe_2_O_3_	CaO	MgO	SO_3_	K_2_O	Na_2_O	Na_2_O_e_ ^(a)^	LOI
Cement	17.42	4.30	3.30	66.17	1.45	3.33	1.21	0.46	1.26	2.35
Fly ash	51.45	26.00	7.65	3.58	1.71	0.93	3.84	<0.10	<2.63	4.85

^(a)^ Available alkali, expressed as Na_2_O_e_, as per ASTM C311.

**Table 3 materials-13-05135-t003:** Chemical composition of lime, wt % (by X-ray fluorescence).

	Ca(OH)_2_	SiO_2_	Al_2_O_3_	Fe_2_O_3_	MnO_2_	MgO	CaCO_3_
Lime	92.0	0.6	0.3	0.2	0.01	0.7	6.0

**Table 4 materials-13-05135-t004:** Total porosity (%).

Age	Mix			
	C-0	CFA-0L	CFA-10L	CFA-20L
28 days	9.4	12.1	11.4	11.1
180 days	9.0	11.5	10.3	10.0

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
