# Peer review of "Influence of Hydrated Lime on the Chloride-Induced Reinforcement Corrosion in Eco-Efficient Concretes Made with High-Volume Fly Ash"

_materials, 2020, doi:10.3390/ma13225135_

Round 1

Reviewer 1 Report

The paper talks about the infuence of hydrated lime on concrete formulations with fly ash. To interpret the results properly, they are missing important information.

First, I am surprised by the concrete formulations where I see a small mass of gravel, but more often the gravel is more present than the sand? Is this formulation standardized?

You have chosen a W / B of 0.5, I have not seen any information on concrete subsidence.

Does fly ash or hydrated lime affect workability?

Have you done a test to find out the% of occluded air?

In addition, lime tends to delay setting, it would have been interesting to make the start and end of sets. During compressive strengths, how many test pieces did you break? I read a values ​​for C-0 of more than 60 MPa at 28 days, I find this value important, what was the desired resistance?

You have made a lot of measurements on devices, it lacks a synthesis and a cross-checking of the results. We have the impression of having a test report.

Tables 1, Figure 9, 12 are misplaced. Figure 1 (a and b) is of poor quality

To finish, review and complete the introduction and conclusion

Reviewer 2 Report

The article is related to the influence of hydrated lime on the chloride-induced reinforcement corrosion in eco-efficient concretes made with high-volume fly ash. For this purpose several tests were carried out including mercury intrusion porosimetry, chloride migration, accelerated chloride penetration, electrical resistivity and corrosion rate.

  • I agree that some of which have cementitious properties, for example fly ash. Please provide a reference to claim this statement (e.g. https://www.sciencedirect.com/science/article/abs/pii/S0959652617323582),
  • Section 2.1. Please justify why 50% of the cement was replaced by fly ash in the mixes. What about other ranges of replacement?
  • How the chemical composition of cement and fly ash presented in Table 2 was determined? Which method has been used?
  • Please provide the scale bars for photographs showing rebars presented in figure 12 and 13,
  • I would like to see some perspectives in conclusion section.

Reviewer 3 Report

The present study is interesting and written well. However, there is no correlation explained by the authors on compressive strength of concrete and corrosion of steel rebar. Higher compressive strength makes the concrete more compact, less porous which resist the ingress of chloride ion towards steel rebar result in higher corrosion resistance while in present study, it is controversial.

If concrete is compact and compressive strength higher then porosity and chloride diffusion would be less while it is vice-versa in present study.

Author shave shown in cleaned rebar surface photo in Fig. 12 while in 13, they have not cleaned with cleaning solution.

Authors must show the cleaned rebar images in Fig. 13. It is better to show the evolution of the corrosion rate result in inset for lower current value which make more clear for authors.

Round 2

Reviewer 1 Report

Thank you for these improvements. I accept the corrections made.

Reviewer 2 Report

The manuscript can be accepted

Reviewer 3 Report

The authors have explained my query very well as well as improved the manuscript. Thus, now it can be accepted for publication.